# Efficient and scalable Path Integral Monte Carlo Simulations with worm-type updates for Bose-Hubbard and $XXZ$ models

N. Sadoune[1,2], L. Pollet[1,2*],

**1** Arnold Sommerfeld Center for Theoretical Physics, University of Munich, Theresienstr. 37, 80333 München, Germany
**2** Munich Center for Quantum Science and Technology (MCQST), Schellingstr. 4, 80799 München, Germany
* Lode.Pollet@lmu.de

October 3, 2022

## Abstract

We present a novel and open-source implementation of the worm algorithm, which is an algorithm to simulate Bose-Hubbard and sign-positive spin models using a path-integral representation of the partition function. The code can deal with arbitrary lattice structures and assumes spin-exchange terms, or bosonic hopping amplitudes, between nearest-neighbor sites, and local or nearest-neighbor interactions of the density-density type. We explicitly demonstrate the near-linear scaling of the algorithm with respect to the system volume and the inverse temperature and analyze the autocorrelation times in the vicinity of a $U(1)$ second order phase transition. The code is written in such a way that extensions to other lattice models as well as closely-related sign-positive models can be done straightforwardly on top of the provided framework.

# 1  Introduction

The worm algorithm is one of the most successful algorithms to simulate sign-positive models. It is based on a path-integral representation of a finite system at finite temperature with Fock states as basis states and a perturbative expansion in the kinetic energy, *i.e.* a strong-coupling expansion with guaranteed series convergence. Unlike diagrammatic techniques formulated directly in the thermodynamic limit, continuous symmetries are never truly broken although any correlation function can approach the thermodynamic correlation function exponentially closely in symmetry-broken phases. We call a worm algorithm an algorithm that provides local updates for the Green's function, usually for the single-particle Green's function. [1] For lattice systems, the imaginary time is treated as a continuous variable [1]. The algorithm was originally introduced almost a quarter century ago by Prokof'ev, Svistunov and Tupitsyn for the one-dimensional Bose-Hubbard model [2]. Compared to the loop algorithm [3, 4] and the stochastic series expansions with (directed) operator loop updates [5, 6] the worm algorithm has the advantage that it is more versatile and can be applied to a wider variety of systems. For soft-core bosonic systems, spin systems with sufficiently large S, or systems with average potential energies that are considerably larger than their average kinetic energies the vast majority of published path integral Monte Carlo

---

[1]Note that some authors call a worm algorithm any Monte Carlo algorithm that deals with a constraint (that could be global, local or topological) by temporarily breaking it.

results in the literature were obtained by the worm algorithm.

The original worm algorithm has been extended to models of classical mechanics [7], multi-particle systems, and bosonic systems in continuous space [8, 9] such as $^4$He. There have been closely related formulations of the worm algorithm to the original one but with different types of updates in the canonical [10, 11] and grand-canonical ensemble [12, 13]. The implementation provided in this work is close to the one of Ref. [14] but differs and is easier to extend (see Sec. 5.8).

Let us review the key argument why the worm algorithm is so successful. Worldlines in a path-integral representation form closed loops in imaginary time due to the properties of the trace operator. For periodic boundary conditions, a superfluid phase will form worldlines with a second type of closed loops, namely those with non-zero winding number along spatial directions. The usual argument is that this topological constraint is overcome in the worm algorithm by sampling the single particle Green's function, which is not subject to it. This can in fact be elaborated, and this is easily illustrated for systems with $U(1)$ symmetry. In the bosonic language, the superfluid phase has off-diagonal long-range order, implying that the integral over the equal-time single-particle density matrix is divergent,

$$\lim_{V \to \infty} \int d^D \mathbf{r} \, G(\mathbf{r}, \tau = 0) \to \infty, \tag{1}$$

where $V$ is the system's volume. In 3D, this integral is proportional to the system's volume and this allows one to define the condensate density as the asymptotic value of the equal-time single-particle density matrix. In lower dimensions, condensation is not possible, and the integral is not proportional to $V$ although it diverges in the superfluid phase. Since the worm algorithm is devised to sample the single-particle Green's function, it must spend most of the simulation time in the regime of large $\mathbf{r}$ in the superfluid phase (also with open boundary conditions) in any dimension, and essentially updates all the particles in one sweep, where one sweep is defined as a sequence $G(\mathbf{r} = 0, \tau = 0) \to \dots \to G(\mathbf{r} \neq 0, \tau \neq 0) \to \dots \to G(\mathbf{r} = 0, \tau = 0)$. As a consequence, one expects autocorrelation times of order one in the superfluid regime, and this is also seen in practice. In the spin language, the superfluid phase corresponds to a ferro-magnetic easy-plane spin ordering.

The emphasis of this paper is on the implementation. Reviews in the literature [15, 16] can give the interested reader a broad overview of the applications and state-of-the art calculations of the worm alogirthm.

## 2   Definitions and Models

The code provided here can be used to sample two different types of models. First, the Bose-Hubbard model can be simulated, wich is defined as

$$H = -\sum_{\langle i,j \rangle} t_{i,j} b_i^\dagger b_j + \text{h.c.} + \sum_i \frac{U_i}{2} n_i(n_i - 1) + \sum_{\langle i,j \rangle} V_{i,j} n_i n_j - \sum_i \mu_i n_i. \tag{2}$$

Here, the sum over $\langle i, j \rangle$ is understood as the sum over all bonds of the lattice between nearest-neighbor sites. The notation h.c. stands for the Hermitian conjugate. The model parameters are the bosonic hopping amplitude $t_{i,j} > 0$, $U_i$ the amplitude of the local density-density interaction on site $i$, $V_{i,j}$ a density-density interaction between nearest-neighbor sites $i$ and $j$, and $\mu_i$ a local chemical potential on site $i$. Bosons on site $i$ are created (annihilated) by the operator $b_i^\dagger$ ($b_i$), with

the standard bosonic commutation relations $[b_i, b_j^\dagger] = \delta_{ij}$ (and zero otherwise), and $n_i = b_i^\dagger b_i$ is the bosonic density operator.

Second, the XXZ-model for spin-$S$ operators can be simulated, which is defined as

$$H = -\sum_{\langle i,j \rangle} \frac{J_{i,j}}{2}(S_i^+ S_j^- + S_i^- S_j^+) + \sum_{\langle i,j \rangle} J_{i,j}^z S_i^z S_j^z - \sum_i h_i S_i^z. \tag{3}$$

Here, $J_{i,j} > 0$ corresponds to in-plane ferromagnetic interactions leading to a sign-positive model, $J_{i,j}^z$ is the amplitude for the $S^z S^z$ interaction between nearest neighbors, and $h_i$ is a local magnetic field along the $z$−direction.

For simplicity of notation, the text written below assumes uniform system parameters (*i.e.*, constant system parameters such as $t, U, V, \mu$ and periodic boundary conditions) even though the code can deal with general site and bond-dependent interaction terms. The implementation is however substantially slower when the parameters are not uniform. Algorithmically, the XXZ and Bose-Hubbard models differ only in the value of the matrix elements of the system parameters. We will for simplicity use the bosonic (or particle) language below unless explicitly written otherwise. Parameters that must be positive are the hopping amplitudes (corresponding to the in-plane spin exchange amplitudes $J_{i,j}$ in the spin models, which must hence be ferromagnetic); the other parameters must only be real.

## 3 Requirements

The current code builds on the ALPSCore libraries [17,18] for scheduling, checkpointing, file input and output, and error evaluation. It is therefore required to have ALPSCore installed with C++-14 compiler options.

## 4 Algorithm and Monte Carlo updates

### 4.1 Perturbative expansion

It is convenient to write the Hamiltonian as

$$H = H_0 - H_1, \tag{4}$$

where $H_0$ is diagonal in the computational basis and $H_1$ causes a transition from one basis to state to another. Specifically, the computational basis is the basis of Fock states defined as the set of all occupation numbers on each lattice site, $\{|n_1, \ldots n_{N_s}\rangle\}$, where $N_s$ is the total number of sites and $n_j = 0, 1, \ldots$ can take any positive integer value (the user can also impose a sharp cutoff) on site $j$. The action of the chemical potential and potential energy terms is diagonal with respect to this basis, and these terms belong therefore to $H_0$. The action of a hopping term in $H_1$ is to change one basis state to a definite other one (there is no branching).

Path Integral Monte Carlo methods are formulated for lattices of finite extension and finite temperature $T = 1/\beta > 0$. The central object is the perturbative time-ordered expansion of the

partition function in continuous imaginary time,

$$Z = \text{Tr } e^{-\beta H_0} \sum_{n=0}^{\infty} \int_0^{\beta} d\tau_1 \int_0^{\tau_1} d\tau_2 \ldots \int_0^{\tau_{n-1}} d\tau_n \, H_1(\tau_1) \ldots H_1(\tau_n). \tag{5}$$

Here, the trace is taken with respect to all states in the computational basis. The positivity of this expansion is necessary and requires $t > 0$ (the constants in front of the bosonic hopping operators must be negative on non-bipartite lattices).

The Heisenberg operators are defined as

$$H_1(\tau_j) = e^{\tau_j H_0} H_1 e^{-\tau_j H_0}. \tag{6}$$

The central quantity of interest in the worm algorithm is the single-particle Green's function $G(A, \tau_A; B, \tau_B)$ defined as the thermal average $G(A, \tau_A; B, \tau_B) = \frac{1}{Z} \langle \mathcal{T}[b_A(\tau_A) b_B^\dagger(\tau_B)] \rangle$, where $\mathcal{T}$ is the time-ordering operator. These extra operators in the Green's function sector are referred to as worm operators. The average describes the thermal properties of the propagation of a particle created at site $B$ at time $\tau_B$ and annihilated again at site $A$ at time $\tau_A$ when $\tau_A > \tau_B$. The other time ordering implies the propagation of a particle that is first removed from the equilibrium state and reinserted at a later time.

In order to compute this quantity we sample an extended space $Z_{\text{MC}}$ with

$$Z_{\text{MC}} = Z + C_W Z_G \tag{7}$$

$$Z_G = \sum_{A,B} \int d\tau_A \int d\tau_B \, G(A, \tau_A; B, \tau_B), \tag{8}$$

where $C_W$ is a constant controlling the relative statistics between the partition function sector $Z$ and the Green's function sector $Z_G$, for which we also use a perturbative time-ordered expansion in continuous imaginary time,

$$\text{Tr} \sum_{A,B} e^{-\beta H_0} \sum_{n=0}^{\infty} \int_0^{\beta} d\tau_1 \ldots \int_0^{\beta} d\tau_n \int_0^{\beta} d\tau_A \int_0^{\beta} d\tau_B \frac{1}{n!} \mathcal{T}[H_1(\tau_1) \ldots H_1(\tau_n) b_A(\tau_A) b_B^\dagger(\tau_B)]. \tag{9}$$

Inserting complete sets of computational basis states before and after every non-diagonal operator yields configurations as illustrated in Fig. 1.

The algorithm proposes to change the configurations in the partition function sector by performing local updates in the Green's function sector, and switching between the two sectors — *i.e.*, between the two terms on the right hand side of Eq. 7 — when the worm operators are at the same time and place. The latter is done by the update pair INSERTWORM-GLUEWORM. Updates in the Green's function sector leave one of the worm operators stationary and move the other one around, thereby providing an unbiased sampling of the single particle Green's function. In the update MOVEWORM one of the worm operators can move forward or backward in imaginary time, whereas in the update-pair INSERTKINK-DELETEKINK the worm operator can jump to a neighboring site by creating (removing) a hopping term.

These updates are discussed in more detail below, followed by the Monte Carlo estimators for common observables.

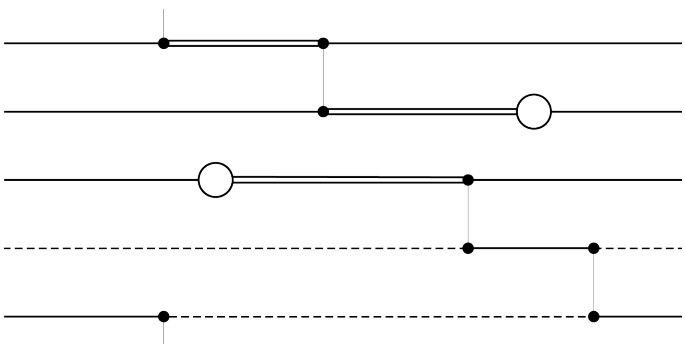

Figure 1: Graphical representation of a typical configuration in the Green's function sector. Imaginary time goes from left to right in the figure and there are five sites. World lines are denoted by single lines (site is once occupied), double lines (site has occupancy two) or dashed lines (site is not occupied). Interactions (hopping of a particle) are denoted by vertical lines. The two circles mark a discontinuity in the world lines and correspond to the worm operators. One of them creates an extra particle, the other one annihilates it. As a consequence of the U(1) symmetry, the total number of particles is conserved at every interaction.

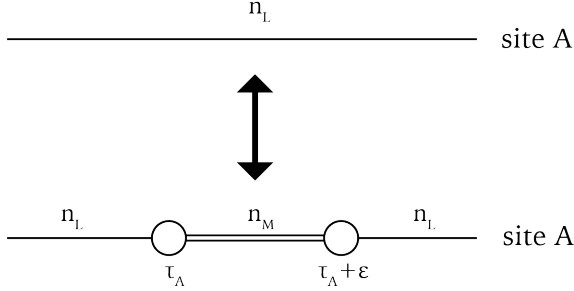

Figure 2: Graphical illustration of the insertion (or deletion) of a worm pair. An arbitrary site $A$ and an arbitrary time $\tau_A$ are chosen. We have shown here the case that the occupation between the worm operators is increased by one.

## 4.2 The INSERTWORM-GLUEWORM update pair

Switching between the partition function sector and the Green's function sector is accomplished by inserting or removing a worm pair. These updates are each other's complement and illustrated in Fig. 2.

INSERTWORM can only be called in the partition function sector. The weights before and after the updates are

$$
\begin{aligned}
W(X) &= \langle n_L | n_L \rangle = 1 \\
W(Y) &= C'_W \langle n_L | b_A | n_M \rangle \langle n_M | b_A^\dagger | n_L \rangle \, d\tau,
\end{aligned}
\tag{10}
$$

where $C'_W$ is a constant controlling the relative statistics between the partition and Green's function sectors. A worm pair is inserted by uniformly choosing a random time $\tau_A$ and a random site $A$. For a homogeneous system this is a natural choice. It is straightforward to implement other rules for non-homogeneous systems. The occupation $n_M$ differs from the existing occupation $n_L$ by $\pm$

1 depending on the chronological order of the $b_A$ and $b_A^\dagger$ operators, which differ in time by an infinitesimal amount. We pick with equal probability between these two choices, and reject the update in case the occupancy $n_M$ is outside the allowed occupancy bounds (say when a negative value would arise for a soft-core Bose-Hubbard model).

Thus

$$P(X \to Y) = \frac{d\tau}{2\beta V} \delta_{A,B} \delta(\tau_A - \tau_B). \tag{11}$$

For the reverse update, we choose to always glue a worm pair together when possible (*i.e.*, when they are infinitesimally close to each other in time, on the same site, and the exterior occupancy $n_L$ is the same),

$$P(Y \to X) = \delta_{A,B} \delta(\tau_A - \tau_B). \tag{12}$$

It follows that if we choose

$$C_W' = \frac{C_W}{\beta V}, \tag{13}$$

we have cancelled all extensive factors and have the correct normalization for the single particle Green's function (see Sec. 4.5). The acceptance factor $r$ for the INSERTWORM update is hence

$$r = \frac{W(Y)P(Y \to X)}{W(X)P(X \to Y)} = 2C_W \langle n_L| b_A |n_M\rangle \langle n_M| b_A^\dagger |n_L\rangle, \tag{14}$$

and the update is accepted with probability $\min(1, r)$ according to the Metropolis-Hastings algorithm [19, 20]. The acceptance factor for GLUEWORM is then naturally $1/r$. The parameter $C_W$ can be set by the user in order to optimize the acceptance factors for both updates based on typical values for the matrixelements $(\langle n_L| b_A |n_M\rangle)^2$.

## 4.3 The MOVEWORM update

During the worm updates, we make the choice, with equal probability, to move only one of the worm operators around in space and time, and call this operator the worm head, the other one is called the worm tail. The efficiency objective of the MOVEWORM update is to cancel the exponential factors found in the configuration weights — which are of the form $\exp(-E\Delta\tau)$ obtained by plugging Eq. 6 into Eq. 5 — through drawing random numbers $\Delta\tau$ from an exponential distribution, $p_{\exp}(\Delta\tau)d\tau = E\exp(-E\Delta\tau)d\tau$ (with $E > 0$). The MOVEWORM update, which is its own reverse, proceeds as follows. First, we select with probability $1/2$ whether the worm head moves forward or backward in imaginary time and determine the imaginary time interval over which the local potential energy stays constant. The boundary of this interval is found by comparing the times of the next (or, previous, in case the worm head moves backward) 'element' on the site of the worm head and its neighbors and choosing the smallest time interval $\tau_{\mathrm{intv}}$ among them. Possible 'elements' are a hopping term, measuring vertices, the worm tail, e.t.c. If the variable $W_\pm$ is set (it is the variable that resolves ambiguities in the time ordering when the worm time coincides with the one of a physical operator. We refer to Sec. 5.2 for more details. The variable can take three values: $W_\pm = 0$ if the worm is exactly at the time of a physical operator, $W_\pm = 1$ indicates that the worm is located at an infinitesimal time later than the one of a physical operator, and $W_\pm = -1$ indicates an infinitesimal time prior to the time of a physical operator), we reject the update if the proposed movement is in the direction of the existing element. Let us assume that the forward direction is chosen to keep the notation simple. Second, we determine the local potential energies just prior to ($E_L$) and just later than ($E_R$) the worm head (according to Fig. 1 this would be just

to the left(L) and to the right(R) of the worm head). To compute the acceptance factor $r = \frac{r_N}{r_D}$, we stress that it is important to understand that ending the worm movement on a time coinciding with the one from an element is radically different from ending on a free time (and, likewise, starting the worm movement from a time coinciding with an existing element is different from starting the movement from a free time). Namely, if the final time is free (*i.e.*, there is no other element there) then it can be reached in only one way, but if the final time coincides with the one of an existing element then all proposed final times going beyond this time would result in the same final configuration. One must hence integrate over all these equivalent possibilities. The update in case of two free ends is illustrated in Fig. 3. With this in mind, we can now distinguish between two cases:

- Case forward in time, $E_R > E_L$. This means that the movement of the worm reduces the potential energy. We draw an exponential deviate, $p = -\ln(u)/E_{\text{off}}$, where $u$ is uniformly distributed over the interval $u \in [0, 1[$, and $E_{\text{off}} \neq 0$ is an offset energy which should be different from 0 for ergodicity reasons (cf. Ref. [14]) . It is an optimization parameter that the user can set with default value 1. If $\tau_{\text{intv}} > p$ then $r_D = E_{\text{off}}$ otherwise $r_D = 1$. If $W_{\pm}$ is set, then $r_N = 1$, otherwise $r_N = E_R - E_L + E_{\text{off}}$.

- Case forward in time, $E_L > E_R$. This means that the movement of the worm increases the potential energy. We draw an exponential deviate, $p = -\ln(u)/(E_L - E_R + E_{\text{off}})$, where $u$ is again uniformly distributed over the interval $u \in [0, 1[$. If $\tau_{\text{intv}} > p$ then $r_D = E_L - E_R + E_{\text{off}}$, otherwise $r_D = 1$. If $W_{\pm}$ is set, then $r_N = 1$, otherwise $r_N = E_{\text{off}}$.

To prevent the roundoff errors (which are inherent to the double precision format of the continuous time variable) we reject updates in which $p$ is too small or would result in the worm being too close to an existing element. In the code the meaning of too close corresponds to $10^{-15}$ in units of $\beta$, which results in an inevitable but very small discretization error. If the update is accepted according to the Metropolis-Hastings algorithm [19, 20], then the worm head moves forward by an amount $p$ if $p < \tau_{\text{intv}}$, with $W_{\pm}$ set to zero, and the time of the worm head is set equal to the time of the next interaction if $p \geq \tau_{\text{intv}}$ with $W_{\pm}$ set to $-1$. Moving backward in imaginary time is the reverse update from moving forward in time. The numerator and denominator of the acceptance factor can be computed as follows:

- Case backward in time, $E_R > E_L$ : We draw an exponential deviate, $p = -\ln(u)/(E_R + E_L - E_{\text{off}})$, where $u$ is uniformly distributed over the interval $u \in [0, 1[$. If $\tau_{\text{intv}} > p$ then $r_D = E_R + E_L - E_{\text{off}}$, otherwise $r_D = 1$. If $W_{\pm}$ is nonzero, then $r_N = 1$, otherwise $r_N = E_{\text{off}}$.

- Case backward in time, $E_L > E_R$ : We draw an exponential deviate, $p = -\ln(u)/E_{\text{off}}$, where $u$ is uniformly distributed over the interval $u \in [0, 1[$. If $\tau_{\text{intv}} > p$ then $r_D = E_{\text{off}}$, otherwise $r_D = 1$. If $W_{\pm}$ is nonzero, then $r_N = 1$, otherwise $r_N = E_L - E_R + E_{\text{off}}$.

It can be checked that the acceptance factors provided above satisfy detailed balance. As absolute energies cannot matter physically, we worked directly with the energy differences $(E_R - E_L)$ to which we added $E_{\text{off}}$ in order to avoid zero when $E_R = E_L$. Note that the way in which a new time is proposed in the MOVEWORM update is similar to the one in Ref. [14], where it was worked out in more detail.

## 4.4 The INSERTKINK-DELETEKINK update pair

The INSERTKINK update is the one in which a hopping term is inserted and the worm jumps to an adjacent site. It proceeds as follows. Note that the update is immediately rejected if $W_{\pm}$ is

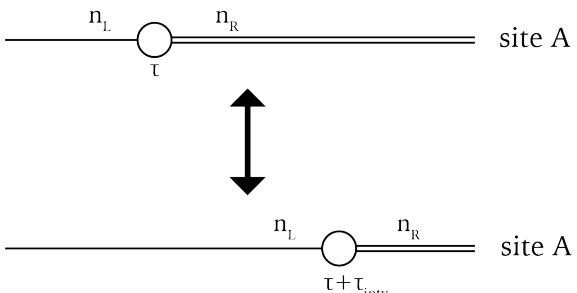

Figure 3: Graphical illustration of the MOVEWORM update. The worm, originally at time $\tau$ is proposed to jump to a later time $\tau + \tau_{\text{intv}}$. The state to the left of the worm is $|n_L\rangle$, with energy $E_L$. The state to the right of the worm is $|n_R\rangle$, with energy $E_R$.

nonzero. A neighboring site $B$ of the worm head site $A$ is chosen with uniform probability, and we must determine the occupation on site $B$ at the time of the worm head. We determine with probability $1/2$ whether the worm should be placed just to the left (infinitesimally prior) or just to the right (infinitesimally later) the newly formed hopping term in imaginary time. Since we consider only models with particle number conservation, the change in particle number on $A$ and $B$ must be opposite, thus the specification of the hopping matrix element is unique. Whenever any of the final occupancies is unphysical we reject the update.

For the reverse update, DELETEKINK, we simply propose to remove the hopping term and let the worm jump to the site linked by the hopping term. This is only possible if $W_\pm$ is nonzero, say $-1$, and the occupancy to the left of the worm is the same as the one to the right of the hopping term on the worm site.

Example for INSERTKINK (see Fig. 4). Let the site on which the worm resides have $z$ neighbors, and we choose one of them with probability $1/z$. The occupation number to the left of the worm head is $n$, to the right of the worm it is $n+1$, and the occupation number on the chosen neighboring site is $m$. Let us place the worm in the new configuration an infinitesimal time later than the newly formed hopping term. Then, for a uniform Bose-Hubbard model, the acceptance factor is

$$r = \frac{2zt\sqrt{m(n+1)}W_{m-1,m}}{W_{n,n+1}}\frac{p_{\text{del}}}{p_{\text{ins}}}, \tag{15}$$

where the worm weights are in the standard choice $W_{m-1,m} = \sqrt{m}$ and $W_{n,n+1} = \sqrt{n+1}$. The probabilities $p_{\text{ins}}$ and $p_{\text{del}}$ are the probabilities with which INSERTKINK and DELETEKINK are picked, respectively.

A situation may occur in which hopping terms cannot be deleted: if, in case the worm is moving from left to right, the occupation to the left of the worm is $n$ and the occupation number to the right of the hopping term is $n\pm2$ (and $n\pm1$ is then the occupation number between the worm head and the hopping term), then the interaction cannot be deleted. It can however be passed by the worm with probability 1 if the standard worm weights are used. This is called PASSINTERACTION in the code.

These few updates guarantee ergodicity and are in practice also efficient enough.

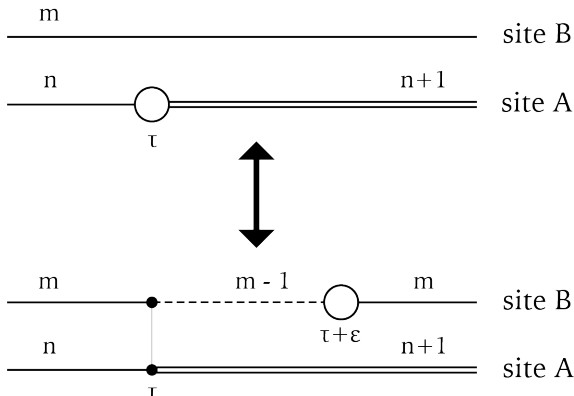

Figure 4: Graphical illustration of the INSERTKINK (or DELETEKINK) update. The worm, originally at time $\tau$ and site $A$, jumps from site $A$ to site $B$, thereby inserting a hopping term between $A$ and $B$. Afterwards the worm moves to an infinitesimally close time $\tau + \epsilon$. Between times $\tau$ and $\tau + \epsilon$ a new state with occupation number $m - 1$ is created. This update is only possible if the occupation number $m$ on site $B$ is larger than zero. Note that in this example $m = n$.

## 4.5 Estimators

Having discussed the set of updates, we now mention the common observables that are implemented.

In the code we introduce on every site measuring elements (called dummies) where we can immediately read off the occupation numbers. Those are conveniently placed at a time $\beta$. In case the user needs to add more diagonal observables, calling the occupation numbers from the dummy iterators is straightforward.

## 4.6 Scalar observables: The total number of particles, kinetic, potential and total energy

The measurement of such observables is cheap. For the kinetic energy, the number of hopping terms $P$ is proportional to the total kinetic energy $E_k = -\frac{\langle P \rangle}{\beta}$. In other words, we count on the fly the number of hopping terms in the simulation, and the measurement is then an $\mathcal{O}(1)$ operation. In practice, the total particle number $\langle N \rangle = \sum_{\mathbf{r}} \langle n(\mathbf{r}) \rangle$ and the potential energy $E_p = \langle H_0 \rangle + \mu \langle N \rangle$ are also updated on the fly and the measurement of these scalar quantities is then also an $\mathcal{O}(1)$ operation. Note that the estimators in the code for the potential and total energy contain the contributions from the chemical potential.

## 4.7 A diagonal vector observable: The density distribution

The density distribution $\langle n(\mathbf{r}) \rangle$ is a vector observable with length corresponding to the total number of sites. It can be determined from knowledge of the occupation numbers at the dummy vertices. It is always implemented as a vector observable in our implementation, also for a system with translational invariance.

## 4.8   The density-density correlation function

The density-density correlation $\left\langle n(\mathbf{r})n(\mathbf{r}')\right\rangle$ function can likewise be computed from knowledge of the occupation numbers at the dummy vertices. The cost of this measurement is however a factor $N_s$ higher than the density distribution and should therefore be called less frequently. Note that such matrix observables are turned off in the code when the system is not uniform. For a translationally invariant system such a matrix observable can be stored as a cheaper vector observable, and this is also done so in our implementation. The option to measure such diagonal matrix observables less frequently is left to the user via the parameter interface (see Sec. 5.5).

## 4.9   Winding numbers

The superfluid density can be related to fluctuations in winding numbers when periodic boundary conditions are used [21]. The code provides an estimator for $\left\langle W^2\right\rangle = \frac{1}{d}\sum_{j=1}^{d}\left\langle W_j^2\right\rangle$ where $d$ is the dimension of the system. The winding number $W_j$ in direction $j$ is updated on the fly during the worm propagation. It is left to the user to relate this quantity to the superfluid density; *e.g.* for a $d$−dimensional cubic lattice the factor $L^{2-d}/\beta$ is not provided in the code.

## 4.10   Equal-time density matrix

The equal-time density matrix $G(\mathbf{r}, 0)$ is histogrammed during the worm updates. Every time the worm head attempts to move across the time of the worm tail, irrespective of its site, we force the worm head to stop at that time. We determine then the distance $\mathbf{r}$ between the sites of the worm head and tail, and have a Monte Carlo estimator $1/C_W$ for $\mathbf{r} = 0$. When the worm head and tail reside on the same time, care must be taken with the non-commutativity of the operators such that a correct density measurement is performed. The equal time density matrix is naturally normalized in our implementation thanks to the detailed balance condition in the INSERTWORM-GLUEWORM updates. Note that the first element of the equal-time density matrix, corresponding to the entry between nearest neighboring sites, is an equivalent measurement of the kinetic energy, up to a minus sign, the total number of bonds, and the value of $t$ for a translationally invariant system.

## 4.11   Green's function in imaginary time

In the canonical framework we provide an implementation of the Green's function at zero momentum as a function of imaginary time, $G(\mathbf{p} = 0, \tau)$, whose histogram can be updated whenever $W_{\pm} = 0$ with the value $1/C_W$ at the entry $\tau$ which is the discretized fractional particle number difference between the situation with and without worms, *i.e.* positive when the worm head is adding particles to the system and negative when the worm head is removing particles. For gapped systems, this quantity will asymptotically behave as $G(\mathbf{p} = 0, \tau) \sim e^{-E_{p/h}|\tau|}$ for $|\tau| \to \infty$, projecting out the energies of a quasi-particle (quasi-hole) excitation for positive (negative) imaginary times, from which the respective dispersions can be obtained. For systems with (emergent low energy) Lorentz symmetry, the (asymptotic) spatial decay of the Green's function must be the same as the temporal decay. This can be another setting where this quantity is useful.

# 5   Implementation

A configuration is completely specified if we know at every site the evolution of the occupancy numbers over imaginary time. For every site we opt to store the information when the occupancy numbers change (note that other implementations work with constant intervals).

## 5.1   Data Structure

The 'element' is the basic building block of the data structure. The data structure on every site must contain all elements in chronological order for which we have chosen the list of the C++standard library. The full configuration then additionally consists of a C++vector over such lists. A hopping term affects the occupancies on two sites and will therefore be encoded as two different elements on two different sites with equal time. The worm operators also induce changes in occupancy, but are local and are also stored as elements on a single site. As mentioned before, we also insert dummy iterators at time $\beta$ where nothing changes but which are used for the Monte Carlo measurement of diagonal observables. The information stored per element are its time, the occupation number to the left and right, the site to which it is linked in case of a hopping term (for measuring elements and the worm operators we link the site to itself for convenience), and the type of event called its color in the code (which is 1 for a hopping term, 0 for a dummy, and -1 for the worm head or tail). Finally, an element contains C++iterators to its so-called associations, which is explained in the next paragraph.

In the Bose-Hubbard model the terms in $U$ and $\mu$ are local, whereas the ones in $t$ and $V$ couple nearest-neighbor sites. In the $XXZ$ model the terms in the magnetic field are on-site and all other ones couple only nearest-neigbor sites. In such cases, one only needs to keep track of the changes in the configuration in the vicinity of the worm head. In order to compute any matrix element between nearest-neighbor sites, it is important that one can quickly lookup the occupation number on the current and its adjacent sites. To ensure that this lookup is an $\mathcal{O}(1)$ operation (*ie*, not scaling with inverse temperature or system size), which is essential if one wants to simulate large lattices in higher dimensions, we store at every element in the configuration a connection (C++iterator) to the next element in the configuration on the nearest neighbors. These iterators are called associations in the code. The meaning of "next element" requires further specification: it is the element on the adjacent site with the earliest time that is equal (eg, as in case of a hopping term) or greater than the current one. This structure needs to be continuously updated during the worm propagation and is the most cumbersome technicality in the code.

We have implemented a few other data structures instead of the list of the C++standard library such as a self-implemented AVL tree, or a self-implemented stack-list, as introduced by J. Greitemann *et al* in Ref. [22] [2]. We found however that these structures offer no advantages over the list of the C++standard library.

## 5.2   Continuous time issues

For completeness, we repeat here the functioning of two parameters in the code that allow us to deal with continuous time variables.

First, since the times are stored in double precision format errors due to the roundoff of floating point numbers can arise. We therefore make sure that hopping terms cannot get closer in imaginary time than a certain value chosen as $10^{-15}\beta$. This introduces a tiny discretization error

---

[2]Note that the AVL tree implementation still uses associations, which is not compatible with a proper tree concept

in the simulation that is for all practical circumstances negligible compared with the statistical uncertainties. Quantities such as the potential energy which are updated on the fly must for the same reason also be recomputed from scratch from time to time.

Second, note that the worm head can move arbitrary close to an element in the list (a hopping term, or a measuring vertex). To ensure the correct chronological order we give the worm then the same time as the element in the list and make use of the variable $W_{\pm}$, which can take three different values: $+1(-1)$, indicating that the worm is at a time infinitesimally later than (prior to) the element, or 0 indicating that the time of the worm is not infinitesimally close to any element. When $W_{\pm}$ is set, the INSERTKINK update is impossible, and MOVEWORM is only possible in the opposite direction.

## 5.3   Lattice definitions

We provide two equivalent lattice implementations, called *static* and *XML*. Depending on requirements, the user might choose one or the other. In both implementations the size of the selected lattice is set during runtime by the parameters Lx,Ly,Lz in the initialization file. Depending on the dimension $d$ of the lattice only the first $d$ length parameters are considered and others ignored. For instance, in case of a square lattice, Lx,Ly are relevant and Lz is ignored. Boundary conditions of the lattice are specified in a similar fashion, that is by providing parameters pbcx,pbcy,pbcz. These correspond to using periodic boundary conditions if the relevant parameter evaluates to true, and open boundary conditions otherwise.

The static lattice implementation is similar to the one of Refs. [23–26]. Lattice selection happens at compile time by specifying the flag LATTICE=<lattice>, thus switching lattice structure requires recompilation (changing the size of the lattice doesn't). If not specified, lattice defaults to chain. All types of lattices are supported in the sense that they can easily be implemented. Currently implemented are chain, square, cubic, honeycomb, triangular, ladder.

The advantage of the XML lattice implementation [27] is that new lattices can be defined without altering the source code. Any types of lattices can be conveniently defined in XML files by specifying basis and unit-cell. Three additional parameters are required in the initialization file: latticefile indicating the path to the XML file containing lattice definitions, basis_name denoting the name of the basis, and cell_name denoting the name of the unit-cell. Winding number computation is currently not available in the XML implementation.

## 5.4   Model definitions

To specify which model to use, the user has to provide a parameter "model" in the initialization file. Currently available are "BoseHubbard" (default) and "XXZ", both with uniform bond and site weights. The uniform parameters of the Bose-Hubbard model in eq. 2 are set by

$$
\begin{aligned}
t_{i,j} &= t: \quad \texttt{t\_hop} \\
U_i &= U: \quad \texttt{U\_on} \\
V_{i,j} &= V: \quad \texttt{V\_nn} \\
\mu_i &= \mu: \quad \texttt{mu}
\end{aligned}
\tag{16}
$$

while the XXZ-model in eq. 3 is determined as

$$
\begin{aligned}
J_{i,j} &= J: & \texttt{Jpm} \\
J_{i,j}^z &= J^z: & \texttt{Jzz} \\
h_i &= h: & \texttt{h}.
\end{aligned} \tag{17}
$$

Additionally the spin of the $XXZ$-model is controled by $2S \equiv \texttt{nmax}$.

## 5.5 Other Simulation parameters

There are a number of other simulation parameters that the user can set:

1. From the Monte Carlo framework in ALPSCore [17,18] we inherit the variables `runtimelimit`, `sweeps`, and `thermalization`. They mean the total runtime in seconds, the number of Monte Carlo sweeps performed during the measuring stage, and the number of thermalization sweeps, respectively.

2. The variable `E_off` was discussed in the MOVEWORM update. It can take any strictly positive value and controls the size of the time jump in the MOVEWORM update. A sensible choice is an estimate of $U \langle n \rangle$ for a uniform system with on-site interactions only.

3. The variable `seed` is the seed of the random number generator. It can be any unsigned integer.

4. The variable `Ntest` performs a number of tests on the validity of the configuration after the indicated number of sweeps. After `Nsave` sweeps the current state of the simulation is checkpointed. These two variables should be large enough such that their computational cost is negligible in comparison with the time spent on sampling.

5. The variable `C_worm` is the parameter $C_W$ that we previously discussed in the INSERTWORM-GLUEWORM update pair. A value of the order 1 is recommendable for default settings of the code.

6. The variables `Nmeasure` and `Nmeasure2` determine after how many sweeps diagonal observables with cost $\mathcal{O}(1)$ and cost $\mathcal{O}(N_s)$ are measured, respectively. These numbers should be large enough such that enough statistics is acquired but small enough such that that their computational cost remains negligible. The first one can be set roughly equal to the autocorrelation time, the second one should typically be a factor $N_s$ (total number of sites) smaller.

7. The user also has the option to modify the probabilities with which the updates are called. The variable names `p_moveworm`, `p_insertkink`, `p_removekink`, and `p_glue` are self-explanatory and correspond to the possible updates that can be called in the Green's function sector. These numbers are normalized to one and in practice one always takes the same probabilities for inserting and removing a kink. In the partition function we have only one update and the corresponding variable `p_insertworm` is therefore set to one.

## 5.6   Optional compiler flags

The user has the possibility to compile the code with a number of flags:

1. The flag `DEBUG` is used when debugging the code and provides a lot of testing and output to the screen. It should certainly be turned off in any production mode.

2. The flag `UNIFORM` assumes constant system parameters in the Hamiltonian. It leads to a significant speed-up for large system sizes due to a significant reduction in memory usage.

3. The user has the option to modify the data structure with the flag `DSTRUC` The default option is the C++list, but one can also choose an AVL tree or the `LIST_STACK` from Ref. [22].

4. The flag `CWINDOW` effectively constrains the simulation to a canonical simulation in the partition function sector. The simulation is initialized with the number of particles specified by the parameter variable `canonical`. Updates in the Green's function sector restrict the separation in imaginary time (but not in space!) between the worm head and the worm tail to a time window specified by the variable `can_window`. This is a hard cutoff in imaginary time without making use of reweighting techniques, *i.e.*, the maximum separation in imaginary time between the worm head and tail in absolute value is this variable times the inverse temperature $\beta$. Note that the proper usage of this flag still requires that the user has set the chemical potential correctly.

Calling the executable with arguments can be done in a number of ways but is inherited from the ALPSCore framework. We refer to the ALPSCore documentation [17, 18].

## 5.7   Output of the code

When the simulation is finished, the serial code provides two `hdf5` files. One `hdf5` file is used for checkpointing (the filename contains the word "clone" in it), *i.e.*, if the simulation has not completed all sweeps yet then it can be continued by adding this file as argument to the executable on the command line, as provided by the ALPSCore framework. The other `hdf5` file contains the output of the simulation (the filename contains the word "out" in it). We further provide a number of tools in order to easily extract such information from the `hdf5` files as the system parameters, the update statistics, and the mean value and error of correlation functions.

## 5.8   Comparison with previous implementations

The present implementation takes over a number of similar ideas from previous implementations of the worm algorithm but differs in some other aspects. Here, we briefly want to mention these similarities and differences.

1. The data structure, the update pair INSERTWORM-GLUEWORM, and the use of exponential deviates (including the variable $E_{\text{off}}$) are (nearly) identical as in Ref. [14] and differ from Ref. [2].

2. Ref. [14] employed the idea of "directed loops", in which the direction of the worm propagation in MOVEWORM is always determined by the previous update. In the current implementation, and in Ref. [2], there is no such notion. With directed loops, detailed balance was broken after the MOVEWORM update and only restored when the worm was halted

as, for instance, when inserting or removing a hopping element. Although the acceptance factors for MOVEWORM and INSERTKINK-DELETEKINK are always 1 in Ref. [14] this is not the case in the present implementation (but it is not a disadvantage for the autocorrelation times).

3. In the present implementation, the worm head is an actual element in the configuration list. It is physically removed and inserted when inserting and deleting a hopping matrix element or when moving the worm around. This was not the case in Ref. [14]. Whereas it may look that the current implementation does more (unnecessary) work, this is actually not true since there are gains when updating the associations in the code that more than compensate the cost, resulting in an overall small gain.

The additional advantages of the current design is that it is simpler to extend the algorithm to multi-species systems, and it also leads to a simpler measurement of the single particle Green function in imaginary time, $\mathcal{G}(\mathbf{r}, \tau)$.

# 6 Scaling of autocorrelation times

In this section we examine the scaling of autocorrelation times in, for simplicity, the Bose-Hubbard model. The unit for the autocorrelation time is one sweep, i.e. one completed worm update from INSERTWORM to GLUEWORM. We start with determining the scaling of the autocorrelation times deep in the superfluid and critical regimes. Below we separately examine the scaling with the total number of sites and inverse temperature. Our observable of choice is the winding number squared along the $x-$ direction because this quantity corresponds to an 'order parameter' for the superfluid phase and is hence expected to couple to the slowest modes in the system.

## 6.1 Scaling of autocorrelation times as a function of system size

We take system parameters such that we are in superfluid regime and increase the system size at fixed temperature. Specifically, we take a Bose-Hubbard model on a square lattice with $t_{\text{hop}} = 1$, $U_{\text{on}} = 12$, $\mu = 4.8$ and fixed $\beta = 10$, which is at the boundary of where the Bogoliubov approximation remains applicable. The result is shown in Fig. 5. The total number of updates in the Green's function sector scales with the total number of sites, and the converged autocorrelation times are nearly constant, too.

## 6.2 Scaling of autocorrelation times as a function of inverse temperature

We take system parameters such that we are in superfluid regime and increase $\beta$ such that we approach the ground state.

As in the previous paragraph, we take a Bose-Hubbard model on a square lattice with $t_{\text{hop}} = 1$, $U_{\text{on}} = 12$, $\mu = 4.8$ but now we choose a fixed systems size $L_x = L_y = 64$. The result is shown in Fig. 6. In accordance with the previous paragraph, we see that the total number of updates in the Green's function sector scales with the inverse temperature, and that the autocorrelation times remain nearly constant, too.

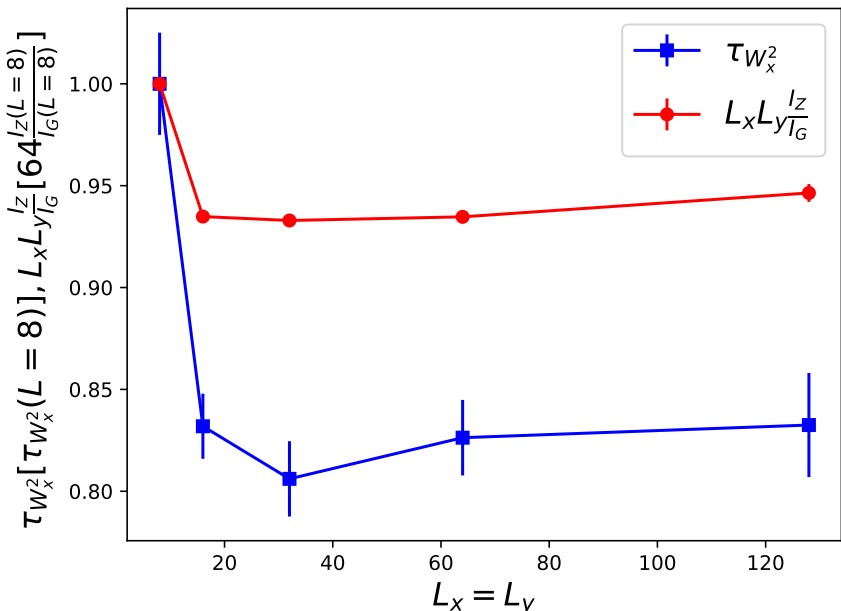

Figure 5: Integrated autocorrelation time of the winding number squared along the $x-$direction (blue line) and the ratio of the number of updates in the partition function to the number of updates in the Green's function sector $\frac{\mathcal{I}_Z}{\mathcal{I}_G}$ multiplied with the system size $L_x L_y$ (red line) as a function of linear system size $L_x = L_y$ for a 2D Bose-Hubbard model in the superfluid regime with parameters $t_{\text{hop}} = 1$, $U_{\text{on}} = 12$, $\mu = 4.8$ and fixed inverse temperature $\beta = 10$. The autocorrelation time and the red full line show nearly constant behavior. Similar scaling behavior is seen for the kinetic energy. The results are normalized to the values found for the smallest system size $L_x = L_y = 8$, for which $\tau_{W_x^2} \approx 55(1)$ and $\frac{\mathcal{I}_Z}{\mathcal{I}_G} \approx 4.574(4) \times 10^{-5}$. Error bars have been determined based on 20 independent runs with different seeds.

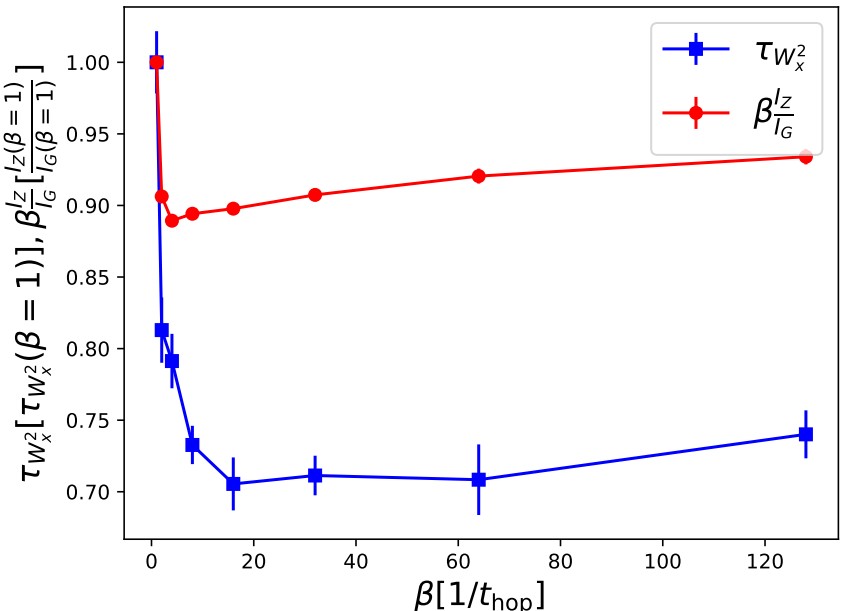

Figure 6: Integrated autocorrelation time of the winding number squared along the $x-$direction (blue line) and the ratio of the number of updates in the partition function to the number of updates in the Green's function sector $\frac{\mathcal{I}_Z}{\mathcal{I}_G}$ multiplied with the inverse temperature $\beta$ (red line) as a function of inverse temperature $\beta$ for a 2D Bose-Hubbard model in the superfluid regime with parameters $t_{hop} = 1$, $U_{on} = 12$, $\mu = 4.8$ and fixed system sizes $L_x = L_y = 64$. The autocorrelation time and the red line show nearly constant behavior. Similar scaling behavior is seen for the kinetic energy. The results are normalized to the values found for the smallest inverse temperature $\beta = 1$, for which we found $\tau_{W_x^2} \approx 59(1)$ and $\frac{\mathcal{I}_Z}{\mathcal{I}_G} \approx 7.54(1) \times 10^{-6}$. Error bars have been determined based on 20 independent runs with different seeds.

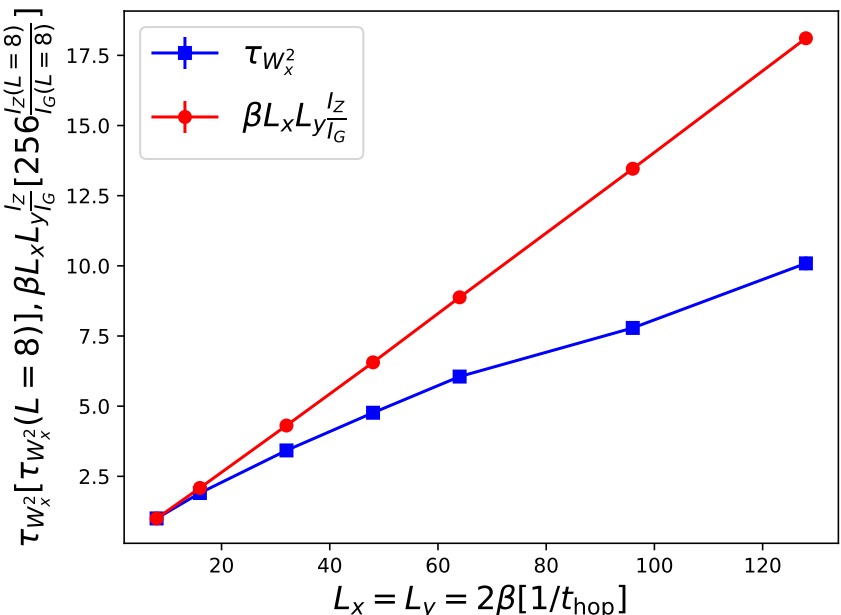

Figure 7: Integrated autocorrelation time of the winding number squared along the $x-$direction (blue line) and the ratio of the number of updates in the partition function to the number of updates in the Green's function sector $\frac{\mathcal{I}_Z}{\mathcal{I}_G}$ multiplied with the system volume $L_x L_y \beta$ (red line) as a function of linear system size for a 2D Bose-Hubbard model in the critical regime with parameters $t_{\text{hop}} = 1$, $U_{\text{on}} = 16.7424$, $\mu = 6.22$ and system sizes $L_x = L_y = 2\beta$. The results are normalized to the values found for the smallest system size, for which we found $\tau_{W_x^2} \approx 133(2)$ and $\frac{\mathcal{I}_Z}{\mathcal{I}_G} \approx 0.000717(1)$. Error bars have been determined based on 20 independent runs with different seeds.

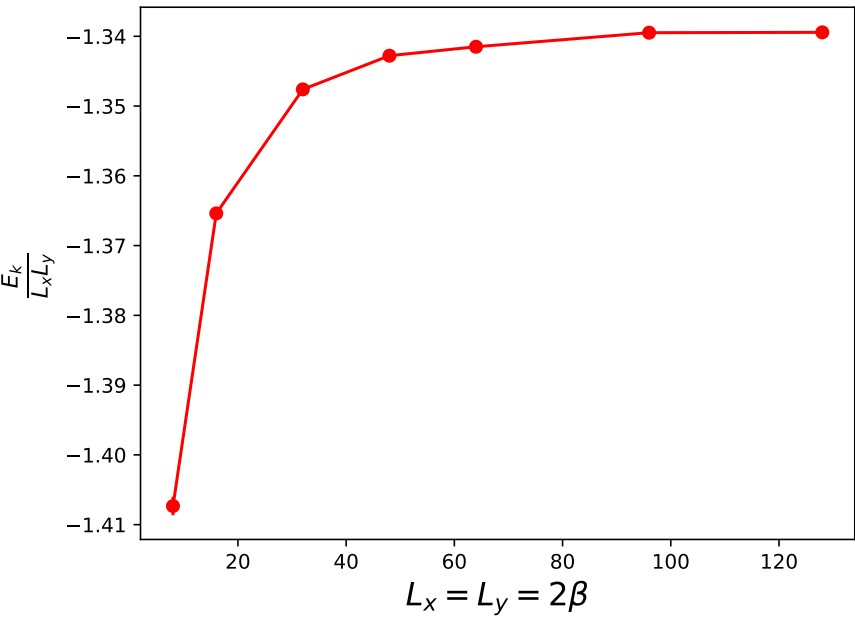

Figure 8: Kinetic Energy per site as a function of linear system size for a a 2D translationally-invariant Bose-Hubbard model on the square lattice with periodic boundary conditions, $t = 1$, $U = 16.7424$, and $\mu = 6.22$ (*i.e.*, at the tip of the Mott lobe with density $n = 1$). The total memory usage is proportional to the kinetic energy via its estimator, $\langle E_k \rangle = -\frac{\langle P \rangle}{\beta}$, where $P$ is the number of hopping elements in the configuration.

## 6.3   Scaling of autocorrelation times in the critical regime

We take system parameters such that we are in the critical regime of the superfluid to Mott insulator regime at constant density. We take a Bose-Hubbard model on a square lattice with $t_{\text{hop}} = 1$, $U_{\text{on}} = 16.7424$, $\mu = 6.22$ and vary $L_x = L_y = 2\beta$ at constant aspect ratio. The growth in the system volume is hence cubic. Note that the quantum critical regime has an emergent conformal theory, hence correlations along spatial and time axes are expected to spread equally. Nevertheless, the reader is warned that thermalizing such a critical system for the biggest system sizes requires a lot of time, and we found it necessary to reset the statistics after 8 CPU hours several times in order to make sure that the statistics are adequate. The result for the autocorrelation time as a function of linear system size is shown in Fig. 7. It clearly grows sublinearly, but the exact behavior is difficult to establish. For wom-type simulations of the classical 3D $XY$ model, a logarithmic growth of the integrated autcorrelation time could be established [7].

## 6.4   Efficiency

Finally, we examine the memory usage and efficiency of our implementation when we go over from a small system to a rather large system. We see in Fig. 8 for a critical 2D Bose-Hubbard model that the kinetic energy per site approaches a constant as a function of linear system size. Since the code stores the hopping events in memory, total memory usage to store the configuration is proportional to the number of hopping events $P$, and thus proportional to two times $|E_k|\beta \sim L_x L_y \beta$. The total average memory consumption can be estimated from the basic data structure which contains 4

integers (which the user can specify), 1 double, and 2$d$ (the coordination number, more generally) C++ iterators. How much memory is required for this data structure is hence lattice, user, compiler, and hardware dependent. Note that the C++ operator `sizeof(Element)` can provide this information. Assuming 4 bytes for an integer, 8 bytes for a double, and 8 bytes for the iterator, then the size of an element is 72 bytes for a cubic lattice and 56 bytes for a square lattice. For the linear system size $L = 96$ in Fig, 8 the average memory usage for storing the configuration is then slightly less than 70 megabytes. Doubling this number to account for fluctuations in kinetic energy gives a realistic estimate for the required memory resources for storing the configuration, excluding the Monte Carlo measurements and smaller overheads. We observe no loss in performance when increasing the system volume, the total number of updates per second is slightly above $2 \times 10^7$ for a thermalized system obtained on a single node of an iMac with a 3.1 GHz Intel Core i5 processor with 24 GB 1667 MHz DDR4 memory.

## 7 Testing

Automated tests for the grand-canonical Bose-Hubbard model as well as the $XXZ$ model with spin values $S = 1/2$ and $S = 1$ are provided within the directory `test_mpi`. To build and run a specific test case, the user can simply execute the provided shell-script `test.sh` for an embarrassingly parallel computation or alternatively `test_serial.sh` for the test to run serial. These scripts take the name of a predefined parameter file from the sub-directory `parameter_files` as argument (without the extension `".ini"`). Once completed, the worm-algorithm results can be compared to the appropriate results from exact diagonalization by executing the python script `compare.py` with the same argument as before.
A possible test run might look as follows:

```
>> cd test_mpi/BoseHubbard_GrandCanonical
>> zsh test.sh chain
>> python3 compare.py chain
```

## 8 Conclusion

In conclusion, we provided a novel open-source implementation of the worm algorithm. The code only relies on a C++-14 compliant compiler and the ALPSCore libraries [17, 18]. The reason for this minimal use of libraries is that we want to keep maintaining the code light and as independent from other software as possible. Our implementation can simulate the Bose-Hubbard model and sign-positive XXZ models on arbitrary lattices with local and nearest-neighbor interactions. Our main data structure is a C++ vector (over the sites) where each vector element contains a C++ list which stores the chronological order implied by the time-ordering operator from many-body field theory.

Our standard lattice implementation is based on Ref. [26] and in our approach the user should construct a different executable for a different lattice type. We have provided instances of a chain, a ladder, a simple square, a triangular and a honeycomb lattice, and a simple cubic lattice. Other types of lattices can straightforwardly be added to the code. Alternatively, lattices based on XML

input files can be used following the library [27]. The current implementation scales well with system size and inverse temperature and can be used on high-performant supercomputers through the ALPSCore libraries.

# Acknowledgements

We wish to thank the ALPS community for many years of scientific collaboration. Our implementation is based on the ALPSCore library [17, 18] and takes two data structures from Ref. [22]. The static lattice implementation is similar to the one of the TK-SVM library [23–26], whereas the XML lattice implementation originates from [27].

The code is available under the GPL license 3.0 at https://github.com/LodePollet/worm.

**Funding information** NS and LP acknowledge support from FP7/ERC Consolidator Grant QSIM-CORR, No. 771891, and the Deutsche Forschungsgemeinschaft (DFG, German Research Foundation) under Germany's Excellence Strategy – EXC-2111 – 390814868.

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
