# Peer review of "Efficient and scalable Path Integral Monte Carlo Simulations with worm-type updates for Bose-Hubbard and XXZ models"

_SciPost Physics Codebases, doi:SciPost Phys. Codebases 9-r1.0 (2022) , SciPost Phys. Codebases 9 (2022)_

## Round 1 · Referee Report · Anonymous (Referee 1) · 2022-5-23

Strengths

This paper and associated code package for the efficient and versatile worm algorithm is a very welcome and useful addition to the slowly growing set of open Quantum Monte Carlo and related packages. A very big thank you from this referee to the authors. The paper is very clearly written and complete (with a few minor additions suggested below), including test cases.

Report

The paper is ideally suited to SciPost. Publication greatly recommended.

Requested changes

A few minor additions to the text would assist the reader:
Page 2: "The implementation provided in this work is close to the one in Ref. [14], but differs...". Since thus there is no published paper with the identical algorithm, the present paper should explain in more detail the differences with respect to Ref.[14]. This reference might also be cited again when $E_{\rm off}$ is introduced.

Figures 5,6,7: The labelling of the vertical axes is misleading. They should explicitely contain a normalization constant, as explained in the captions, where also the unit of time (sweep?) should be specified.

Figure 8: memory in bytes?

It would be helpful to readers trying to find software for their specific simulation needs if pointers to other open Quantum Monte Carlo packages for lattice systems could be included in the introduction (even if that list may be incomplete), for example to other code related to ALPS, or to the ALF code, or a pointer to MateriApps.

  • validity: top
  • significance: top
  • originality: -
  • clarity: top
  • formatting: perfect
  • grammar: perfect

Author:  Lode Pollet  on 2022-09-15  [id 2820]

(in reply to Report 1 on 2022-05-23)

We wish to thank the Referee for their interest in our work, their valuable comments, and their report.

Below we answer to the questions/criticisms raised by the Referee: 1) On "The implementation provided in this work is close to the one in Ref. [14], but differs.." : The current algorithm differs as follows from Ref 14 (now Ref 2 in the new version): (i) The algorithm of Ref[14] was directed, implying that the worm operator keeps propagating in the same direction in the next step as in the current step. This implies that in the MOVEWORM update there is no choice about the direction of propagation to be made. (ii) in the current algorithm, the worm head is an actual element in the configuration list; it is physically removed and inserted when inserting and deleting a hopping matrix element. This was not the case in Ref. [14]. Whereas it may look that the current implementation does more (unnecessary) work, this is actually not true since there are gains when updating the "associations" in the code that more than compensate the previously mentioned cost, resulting in an overall small gain. A second advantage of the present approach is that extensions to multi-worm codes needed for, for instance, multi-species Hamiltonians are simpler. (iii) In Ref.[14] detailed balance is broken after the MOVEWORM update, but this is restored when inserting/removing hopping elements, and results in an acceptance factor 1 throughout the worm propagation. In the present algorithm, detailed balance is respected at all times (but the acceptance factor is not 1 all the time). This has the advantage that quantities such as the single-particle Green function as a function of imaginary time, G(r, \tau), can be measured straightforwardly.

The use of exponential deviates for worm propagation (including the variable E_{off} as the Referee correctly points out and to which we added a citation as requested) is the main similarity with Ref.[14] (and the main difference with the original worm code of Ref. 2 (which was incorrectly cited, but corrected in the resubmitted version) and the old ALPS implementation), and also the INSERTWORM and GLUEWORM updates are the same as in Ref 14.

The sentence "The implementation provided in this work is close to the one in Ref. [14], but differs.." is written very early in the paper where the necessary concepts to explain the difference have not been introduced yet. In order to make the differences with Ref 14 clear, we wrote a new section 5.8, further down the text, and with a full explanation.

2) "Figures 5,6,7: The labelling of the vertical axes is misleading" We have relabelled the figure axes. We replaced "time spent" with "number of updates", because this is the quantity that is actually counted in these measurements.

3) "Figure 8: memory in bytes?"

We have replaced Fig.8 by a new figure in which the kinetic energy per site is shown, which is proportional to the total number of vertices $

$. The latter is proportional to the total memory used to store the Monte Carlo configuration. We have made sure that the quantity has converged for all system sizes shown, and the efficiency of the code is measured with respect to a well-thermalized configuration. We observed no loss in performance, as explained in the revised version.

4) "It would be helpful to readers trying to find software for their specific simulation needs if pointers to other open Quantum Monte Carlo packages for lattice systems could be included in the introduction (even if that list may be incomplete), for example to other code related to ALPS, or to the ALF code, or a pointer to MateriApps." We certainly see the value of having a globally valid library where pointers to open source code are provided for a certain field of research, but we respectfully disagree that this paper is the right place to do so. To give one example, we recently received notification that the old ALPS website is down due to security reasons, and that would make such information useless in a paper. We therefore decided to add links to related Monte Carlo software (ALPS/ALPSCore, ALF, MateriApps, TRIQS, TKSVM, NetKet, ...) directly to our github page, where it is simpler to maintain. On the MateriApps website there is also a link to ALF for instance. Perhaps a system of mutual links to open source software is a more stable system to maintain. We also note that we have implemented a new branch on the Github repository in which compatibility of our worm algorithm with the XML lattice of Synge Todo (https://github.com/todo-group/lattice) is provided, which is nearly fully compatible with the old ALPS lattice class.

---

## Round 1 · Referee Report · Anonymous (Referee 2) · 2022-6-15

Strengths

1- useful contribution to the (still) small set of open source QMC codes 2- detailed explanation of the algorithm

Weaknesses

1- unclear benchmarks 2- worked examples do not refer to physics

Report

This submission is an implementation of the worm Quantum Monte Carlo method capable of efficient simulations of Bose-Hubbard and unfrustrated spin Hamiltonians.

Few to none state-of-the-art implementations of this algorithm are openly available so this is a very useful contribution and should be published. The description of the algorithm is thorough and benchmarks as well as automated tests are provided.

However, in their current state, there are some shortcomings in the user documentation and benchmarks (detailed below) that should be addressed before publication.

Requested changes

1- The presentation of the benchmarks is lacking in different ways. 1a- As mentioned in the main text, it is not clear whether all benchmark simulations are thermalized. Showing such data can misrepresent the actual performance since the underlying Monte Carlo configuration may not have reached its final size during the simulation. The benchmarks should be run for longer or alternatively, the figures should show only system sizes and temperatures without the thermalization problem. 1b- Figs. 5-7 have unrelated data (e.g. autocorrelation times and I_Z//I_G) normalized to 1 at an arbitrary data point. This is misleading and should be replaced either by separate y-axes or by displaying the data in separate panels. 1c- The meaning for I_Z//I_G for performance is not clear. Is it really the ratio of computation time or the ratio of MC steps in the two sectors? Since the nature of the updates depend on the sector, this difference may be significant and should be clarified along with an explanation what the ratio means e.g. for errorbars of observables. 1c- Fig. 8 is missing the unit for the memory. Further, the caption says “function of system volume” but the x-axis label suggests linear system size. The semilog scale hides the expected scaling (linear in βL²?). A linear or log-log scale should be used instead so that the scaling (or its absence) is evident.

2- In the repository, the authors provide the parameter files used to produce the figures as example applications. As a worked example this is not optimal because a user will not want to produce benchmarks but calculate physical observables.

The user guide should contain at least one self-contained example that outlines the step-by-step process of simulating a specific model including how to get a physical observable of interest with errors (using the tools mentioned in 5.7) and the expected result. Currently, this information has to be searched in different places. Putting it together in one place will be especially helpful for nonpractitioners.

3- In the introduction, there is the statement “In particular, soft-core bosonic systems, spin systems with sufficiently large S, or systems with average potential energies that are considerably larger than their average kinetic energies are best simulated by the worm algorithm. The slight loss in efficiency for hard-core bosonic systems or spin-1/2 systems that the worm algorithm displays compared to highly-specialized and optimized algorithms, is in practice irrelevant.” Without a reference to prove these strong claims, they may be a bit unfair to the other available QMC flavors. (a) There exist similarly general formulations of the stochastic series expansion [Phys. Rev. E. 67, 046701 (2003)] able to treat the mentioned systems. Are they slower? (b) It is not clear how much faster specialized methods are and how that difference scales with system size. Since simulations of critical properties require large system sizes, such performance benefits can be relevant in practice.

The claims made in the statement should be either proven with further data or references or they should be removed.

  • validity: top
  • significance: top
  • originality: top
  • clarity: ok
  • formatting: perfect
  • grammar: perfect

Author:  Lode Pollet  on 2022-09-15  [id 2821]

(in reply to Report 2 on 2022-06-15)

We wish to thank the Referee for their interest in our work, their valuable comments, and their report. Below we reply to their comments and criticisms:

1. "1- The presentation of the benchmarks is lacking in different ways. 1a- As mentioned in the main text, it is not clear whether all benchmark simulations are thermalized. Showing such data can misrepresent the actual performance since the underlying Monte Carlo configuration may not have reached its final size during the simulation. The benchmarks should be run for longer or alternatively, the figures should show only system sizes and temperatures without the thermalization problem. "

We understand that the Referee is referring to Fig 8. We have replaced Fig.8 by a new figure in which the kinetic energy per site is shown, which is proportional to the total number of vertices P. The latter is proportional to the total memory used to store the Monte Carlo configuration. We have made sure that the quantity has converged for all system sizes shown, and the efficiency of the code is measured with respect to a well-thermalized configuration. We observed no loss in performance, as explained in the revised version.

"1b - Figs. 5-7 have unrelated data (e.g. autocorrelation times and I_Z / I_G) normalized to 1 at an arbitrary data point. This is misleading and should be replaced either by separate y-axes or by displaying the data in separate panels."

The axes of the figure have been changed to make the normalization of the data clear. We disagree however that the shown quantities are unrelated: the figures show that the constant scaling of the autocorrelation times deep in a phase is explained by the worm exploring a larger system volume in a single sweep. The quantity I_G / I_Z is hence a measure for the "worm length", and the factors of L or beta show the scaling behavior of the worm length.

"1c- The meaning for I_Z / I_G for performance is not clear. Is it really the ratio of computation time or the ratio of MC steps in the two sectors? Since the nature of the updates depend on the sector, this difference may be significant and should be clarified along with an explanation what the ratio means e.g. for errorbars of observables."

As the referee correctly points out, it is the ratio of MC steps in the two sectors and, as mentioned above, a measure for the "worm length". We have improved the wording in the text, and in particular removed the word "time" from this context which leads to a false impression.

"1d- Fig. 8 is missing the unit for the memory. Further, the caption says “function of system volume” but the x-axis label suggests linear system size. The semilog scale hides the expected scaling (linear in βL^²?). A linear or log-log scale should be used instead so that the scaling (or its absence) is evident.

We refer to our answer under issue 1a.

2. "In the repository, the authors provide the parameter files used to produce the figures as example applications. As a worked example this is not optimal because a user will not want to produce benchmarks but calculate physical observables.

The user guide should contain at least one self-contained example that outlines the step-by-step process of simulating a specific model including how to get a physical observable of interest with errors (using the tools mentioned in 5.7) and the expected result. Currently, this information has to be searched in different places. Putting it together in one place will be especially helpful for nonpractitioners."

In light of this remark and a similar one made by the third referee, we added tutorials to the repo. Currently, there are two tutorials: one for calculating the characteristic observables of the normal, superfluid, and Mott insulating phases in the Bose-Hubbard model. The second tutorial compares the Heisenberg model defined on the chain lattice for a spin-1/2 (critical) system with a spin-1 (gapped) system.

3. "In the introduction, there is the statement “In particular, soft-core bosonic systems, spin systems with sufficiently large S, or systems with average potential energies that are considerably larger than their average kinetic energies are best simulated by the worm algorithm. The slight loss in efficiency for hard-core bosonic systems or spin-1/2 systems that the worm algorithm displays compared to highly-specialized and optimized algorithms, is in practice irrelevant.” Without a reference to prove these strong claims, they may be a bit unfair to the other available QMC flavors. (a) There exist similarly general formulations of the stochastic series expansion [Phys. Rev. E. 67, 046701 (2003)] able to treat the mentioned systems. Are they slower? (b) It is not clear how much faster specialized methods are and how that difference scales with system size. Since simulations of critical properties require large system sizes, such performance benefits can be relevant in practice.

The claims made in the statement should be either proven with further data or references or they should be removed."

  1. The Referee is certainly right to point at [Phys. Rev. E. 67, 046701 (2003)] where a general formulation of the stochastic series expansion (SSE) method is presented capable of dealing with long(er)-range interactions. This was used successfully for instance in K. R. Fratus and M. Srednicki, arXiv:1611.03992. An SSE-like algorithm was also used more recently for Rydberg systems, see eg https://arxiv.org/abs/2107.00766, and a study of the same Rydberg system based on worm updates was published in Physical Review B 105, 174417 (2022). One sees that both implementations suffer from slowing down near the phase transitions in a very similar fashion. The autocorrelation times are not mentioned explicitly in the latter paper however. As far as we know, the vast majority of the literature for soft-core particles uses the worm algorithm.

  2. Absolutely. If the goal is to study criticality for a known system and transition, then highly optimized algorithms are called for that give the best possible performance. The purpose of our code by contrast is to make available an implementation that is flexible and good in all sign-free circumstances while perhaps not being the best for certain specific parameter points. There is however evidence in the literature that the scaling of the autocorrelation time is identical within the family of worm-like algorithms, at least for critical Ising models (but the prefactor may differ indeed, and this can be relevant in practice), see N.P. Prokof'ev and B.V. Svistunov, PRL 87, 160601 (2001) for the classical worm algorithm, H. Suwa and S. Todo, PRL 105, 120603 (2010) for a worm algorithm without detailed balance (directed), E. M. Elci et al, PRE 97, 042126 for lifting, and H. Suwa, arXiv preprint 2206.13881 for a combination of directedness and lifting.

As we don't want to add the SSE algorithm to the repo or to this paper and since we are unaware of a published paper that directly compares the two methods, we modified the text as follows: For soft-core bosonic systems, spin systems with sufficiently large S, or systems with average potential energies that are considerably larger than their average kinetic energies the vast majority of published path integral Monte Carlo results in the literature were obtained by the worm algorithm.

---

## Round 1 · Referee Report · Anonymous (Referee 3) · 2022-6-17

Strengths

1-A modern and efficient implementation of an advanced QMC algorithm for Bose-Hubbard and XXZ models is provided
2-A detailed explanation of the algorithmic is provided

Weaknesses

1-Minor presentation issues (see below).
2-Only few references to the actual code are provided.
3-A throughout discussion of physical test cases is missing.

Report

This is a highly welcome contribution, providing us with a modern implementation of the worm algorithm for Bose-Hubbard and XXZ models with quite some flexibility. The algorithmic ideas are explained well, and a few explanations of the implementation are provided. A few test codes and some benchmarks are included. Overall, this paper fits well within the scope of this journal, but a few changes are required (as given below).

Requested changes

1-The explanations in Sec. 4.3 are comparably dense and can easily be improved: (i) Explain, what is meant by "If the variable W_\pm is set", i.e., which values are actually referred to here. (ii) Split up the sentence "To compute the acceptance factor..." into its two separate parts.

2-Some of the text in Sec. 4 speaks about "below" and "above" but the figures are more about "left" and "right" for the imaginary time direction. Please make this notion consistent.

3-It would be useful, if in Sec. 4 more direct references could be made to actual code functions.

4-Split Fig. 5,6, and 7 into two parts (top/bottom) each, one for tau, one for the ratio.

5-"memory" is missing units in Fig. 8. Showing non-thermalized MC data is a no-go, either remove them or simply run those larger-scale simulations sufficiently long (the quoted "15 min. limit" does not really look like somebody did do extensive testing here).

6-One of the test cases (better even one for XXZ and one for Bose-Hubbard) should be worked out and explained in more detail in Sec. 7, instead of just referring to the code base.

7- The text needs to be carefully proofread by the authors and a few remaining language flaws should be eliminated. This applies in particular to the references, such as not using capitals at various places (e.g. "monte carlo"), cf. also the title of Ref. 22.

  • validity: top
  • significance: top
  • originality: -
  • clarity: high
  • formatting: perfect
  • grammar: excellent

Author:  Lode Pollet  on 2022-09-15  [id 2822]

(in reply to Report 3 on 2022-06-17)

We wish to thank the Referee for their interest in our work, their valuable comments, and their report.
Below we reply to their comments and criticisms:

(a) "The explanations in Sec. 4.3 are comparably dense and can easily be improved: (i) Explain, what is meant by "If the variable W_\pm is set", i.e., which values are actually referred to here. (ii) Split up the sentence "To compute the acceptance factor..." into its two separate parts. "

We note that Sec. 4.3 is very close to the corresponding update in Ref 14 (or Ref 2 in the revised version), where it was explained in great detail. We made this clearer in the text. We have further clarified the meaning of the variable W_{\pm}, and split the requested sentence into two separate parts.

(b) "Some of the text in Sec. 4 speaks about 'below' and 'above' but the figures are more about 'left' and 'right' for the imaginary time direction. Please make this notion consistent."

We thank the referee for pointing out this inconsistency. Indeed, Fig.1 shows that imaginary time goes from left to right. We changed all incidences of 'before' and 'above' into 'left' and 'right'.

(c) "It would be useful, if in Sec. 4 more direct references could be made to actual code functions. "

It was our deliberate choice to keep such references to a minimum in the text because in case the code gets updated, or requires maintenance, it may make the text incorrect. We prefer to keep it the way it is. We also note that we added additional tutorials to the repo that should help the reader understand the program better.

(d) Split Fig. 5,6, and 7 into two parts (top/bottom) each, one for tau, one for the ratio.

We believe that this would be counterproductive. The purpose of the figure is to show the same scaling behavior which would be lost in case the figure is split. Fig. 5 shows that the autocorrelation time for the order parameter is independent of system size, and that the number of updates in the Green function sector (as a measure for the "worm length") scales with the system volume (Lx * Ly) to accomplish that. This is indeed the expected behavior (at least for a non-critical system).

(e) "memory" is missing units in Fig. 8. Showing non-thermalized MC data is a no-go, either remove them or simply run those larger-scale simulations sufficiently long (the quoted "15 min. limit" does not really look like somebody did do extensive testing here).

We have replaced Fig.8 by a new figure in which the kinetic energy per site is shown, which is proportional to the total number of vertices P. The latter is proportional to the total memory used to store the Monte Carlo configuration. We have made sure that the quantity has converged for all system sizes shown, and the efficiency of the code is measured with respect to a well-thermalized configuration. We observed no loss in performance, as explained in the revised version.

(f) "One of the test cases (better even one for XXZ and one for Bose-Hubbard) should be worked out and explained in more detail in Sec. 7, instead of just referring to the code base."

In response to this request, and also one by Referee 2, we added two tutorials in the repository: one for the XXZ model, and one for the Bose-Hubbard model. For the XXZ model, we compare the correlation functions of the spin-1/2 Heisenberg model on a chain lattice, which is in a critical phase, with the correlation functions of the spin-1 Heisenberg model on a chain lattice, which is a system with a gap. For the Bose-Hubbard model we compare the distinguishing properties of the normal phase, the superfluid, and the Mott insulator on a square lattice.

(g) "The text needs to be carefully proofread by the authors and a few remaining language flaws should be eliminated. This applies in particular to the references, such as not using capitals at various places (e.g. "monte carlo"), cf. also the title of Ref. 22."

The references were automatically generated from the Bibtex entries provided by the respective journals and the Scipost Bibtex Stylefile. It is the latter that is responsible for removing the capitals in the titles of the references. However, we made sure that nouns such as "Monte Carlo" and "Fröhlich" remain capitalized in the revised version.

---

## Round 2 · Referee Report · Anonymous (Referee 1) · 2022-9-20

Strengths

Strengths as mentioned before. Greatly appreciated.
I am happy with the responses and changes, except for very minor requested additions, see below.

Weaknesses

See requested changes.

Report

Publish.

Requested changes

Very minor. 1) In Figs 5-7, the unit for the autocorrelation time, e.g. "tau_W^2=55" apparently still needs to be specified (updates?). 2) Fig. 8: It would be helpful to roughly know the proportionality factor, or alternatively the actual memory consumption for some system size. 3) Updates per second in the text: hardware should again be specified, similar to v1.

---

## Round 2 · Referee Report · Anonymous (Referee 2) · 2022-9-27

Report

The authors have addressed all of my concerns. I recommend publication as is.

---

## Round 2 · Referee Report · Anonymous (Referee 3) · 2022-9-28

Report

The authors have responded appropriately to the requests. I have no further requests and recommend the revised version for publication.

---

## Round 2 · Author Response

We answered to the Referee Reports separately. All information is provided in our Comments to their Reports.

---

## Round 2 · List of Changes

1. Changed wording throughout the text in response to the Referee Reports; in particular, we replaced Fig 8 with a new figure showing the average kinetic energy per site as a function of linear system size for a critical system.
  2. Added two tutorials to the repo
  3. Added a second way to construct lattices based on XML files. Forked from the library provided by S. Todo.

---

## Round 3 · Author Response

We thank all Referees for their feedback on our paper.
The first Referee made some additional remarks, with which we agree and address below. We thank this Referee in particular for their renewed careful review of our paper.
We hope that our manuscript is now ready for publication.

(i) In Figs 5-7, the unit for the autocorrelation time, e.g. "tau_W^2=55" apparently still needs to be specified (updates?).

Reply:
We added the following sentence to the text:
The unit for the autocorrelation time is one sweep, {i.e.} one completed worm update from INSERTWORM to GLUEWORM.

(ii) Fig. 8: It would be helpful to roughly know the proportionality factor, or alternatively the actual memory consumption for some system size.

Reply: We changed the text as follows:
The total average memory consumption can be estimated from the basic data structure which contains 4 integers (which the user can specify), 1 double, and $2d$ (the coordination number, more generally) \texttt{C++} iterators. How much memory is required for this data structure is hence lattice, user, compiler, and hardware dependent. Note that the \texttt{C++} operator \texttt{sizeof(Element)} can provide this information. Assuming 4 bytes for an integer, 8 bytes for a double, and 8 bytes for the iterator, then the size of an element is 72 bytes for a cubic lattice and 56 bytes for a square lattice.
For the linear system size $L=96$ in Fig,~\ref{fig:efficiency} the average memory usage for storing the configuration is then slightly less than 70 megabytes. Doubling this number to account for fluctuations in kinetic energy gives a realistic estimate for the required memory resources for storing the configuration, excluding the Monte Carlo measurements and smaller overheads.

(iii) Updates per second in the text: hardware should again be specified, similar to v1.
Reply: we modified the text as follows:

We observe no loss in performance when increasing the system volume, the total number of updates per second is slightly above $2 \times 10^7$ for a thermalized system obtained on a single node of an iMac with a 3.1 GHz Intel Core i5 processor with 24 GB 1667 MHz DDR4 memory.

---

## Round 3 · List of Changes

see (i), (ii), (iii) above

---

## Editorial Decision

published